

# Partitioning the impact of environment and spatial structure on alpha and beta components of taxonomic, functional, and phylogenetic diversity in European ants

Xavier Arnan[1,2], Xim Cerdá[3] and Javier Retana[2,4]

[1] Departamento de Botânica, Universidade Federal de Pernambuco, Recife Pernambuco, Brazil
[2] CREAF, Cerdanyola del Vallès Catalunya, Spain
[3] Estación Biológica de Doñana, CSIC, Sevilla, Spain
[4] Univ Autonoma Barcelona, Cerdanyola del Valles Catalunya, Spain

Corresponding author
Xavier Arnan,
xavi.arnan@gmail.com

## ABSTRACT

We analyze the relative contribution of environmental and spatial variables to the alpha and beta components of taxonomic (TD), phylogenetic (PD), and functional (FD) diversity in ant communities found along different climate and anthropogenic disturbance gradients across western and central Europe, in order to assess the mechanisms structuring ant biodiversity. To this aim we calculated alpha and beta TD, PD, and FD for 349 ant communities, which included a total of 155 ant species; we examined 10 functional traits and phylogenetic relatedness. Variation partitioning was used to examine how much variation in ant diversity was explained by environmental and spatial variables. Autocorrelation in diversity measures and each trait's phylogenetic signal were also analyzed. We found strong autocorrelation in diversity measures. Both environmental and spatial variables significantly contributed to variation in TD, PD, and FD at both alpha and beta scales; spatial structure had the larger influence. The different facets of diversity showed similar patterns along environmental gradients. Environment explained a much larger percentage of variation in FD than in TD or PD. All traits demonstrated strong phylogenetic signals. Our results indicate that environmental filtering and dispersal limitations structure all types of diversity in ant communities. Strong dispersal limitations appear to have led to clustering of TD, PD, and FD in western and central Europe, probably because different historical and evolutionary processes generated different pools of species. Remarkably, these three facets of diversity showed parallel patterns along environmental gradients. Trait-mediated species sorting and niche conservatism appear to structure ant diversity, as evidenced by the fact that more variation was explained for FD and that all traits had strong phylogenetic signals. Since environmental variables explained much more variation in FD than in PD, functional diversity should be a better indicator of community assembly processes than phylogenetic diversity.

## INTRODUCTION

A central goal in ecology is to describe patterns of species diversity and composition along broad environmental gradients and to identify the mechanisms that underlie them (e.g., *Pianka, 1966*; *Gaston, 1996*). For instance, broad-scale patterns of species richness are often correlated with contemporary climate (*Gaston, 1996*). However, since climatic factors are usually correlated with latitude, and latitude is, in turn, correlated with other factors, the underlying causes of variation in species richness are much debated (e.g., *Pianka, 1966*; *Willig, Kaufman & Stevens, 2003*). Moreover, other environmental factors besides (or in addition to) climate can affect diversity patterns. For instance, species richness and composition may vary along disturbance (*Fox, 2013*) or habitat heterogeneity gradients (*Rahbek et al., 2007*).

Spatial factors might also influence diversity patterns. In fact, environmental gradients are themselves spatially structured (*Legendre & Legendre, 1998*), and random but spatially limited dispersal of species (*Tuomisto, Ruokolainen & Yli-Halla, 2003*) can also generate spatially structured patterns. Consequently, dispersal limitations and habitat and environmental similarities may result in positive spatial autocorrelation in communities (*Legendre et al., 2009*). In particular, the extent to which species diversity patterns are determined by environmental filters versus random but spatially autocorrelated dispersal are intensely debated (e.g., *Tuomisto, Ruokolainen & Yli-Halla, 2003*). It has been suggested that simultaneously examining the influence of environmental and spatial factors on communities could reveal their relative importance (e.g., *Borcard et al., 2004*). If species diversity patterns solely vary along environmental gradients, it would indicate that the underlying mechanism is environmental filtering; if only spatial structure has an effect, variation in diversity patterns may arise from dispersal limitations. Although there has been a recent increase in the number of studies analyzing the relative contribution of environmental and spatial factors to species diversity patterns, taxonomic diversity (TD) has been the main focus; other diversity components have only rarely been examined (but see *Meynard et al., 2011*).

The problem is that measures of TD treat all species as evolutionarily independent and ecologically equivalent and therefore may not provide enough information regarding the mechanisms underlying community patterns. For this reason, new biodiversity metrics that incorporate information about the functional and phylogenetic characteristics of communities have recently been proposed. While phylogenetic diversity (PD) reflects the accumulated evolutionary history of a community (*Webb et al., 2002*), functional diversity (FD) reflects the diversity of morphological, physiological, and ecological traits found therein (*Petchey & Gaston, 2006*). Understanding how PD and FD relate to TD can provide insights into the extent to which community assembly is driven by deterministic versus stochastic processes (*Cavender-Bares et al., 2009*; *Pavoine & Bonsall, 2011*; *Purschke et al., 2013*). For a given phylogeny of available lineages and evolutionary rate for functional traits, one would expect to see different patterns of phylogenetic and functional community structure depending on whether competition or environmental filtering is the primary driver of community assembly (*Webb et al., 2002*; *Kraft et al., 2007*). It is generally

thought that FD and PD are positively correlated with TD at the regional scale (e.g., *Forest et al., 2007*; *Faith, 2008*; *Meynard et al., 2011*; but see *Losos, 2008*; *Devictor et al., 2010*; *Safi et al., 2011*). If functional traits allow species to locally adapt to environmental conditions (*Pavoine et al., 2011*), it may be that environmental filters predominantly influence the functional structure of communities and that nothing is reflected by their taxonomic and phylogenetic structures (*Díaz et al., 2007*; *Mouchet et al., 2010*). However, a strong correlation between FD and PD would be expected if the functional traits that allow species to persist in the environment are evolutionarily conserved, that is to say, they display phylogenetic signals (*Webb et al., 2002*; *Cavender-Bares et al., 2009*).

Since the processes that shape biodiversity differ across scales (*Whittaker, Willis & Field, 2001*), it is also relevant to study the aforementioned facets of diversity at different scales of analysis (e.g., alpha and beta diversity) (*Devictor et al., 2010*; *Bernard-Verdier et al., 2013*). Analyzing patterns at only one of these scales can result in misleading or incomplete interpretations of the results (*Whittaker, Willis & Field, 2001*). For instance, if environmental determinism is at work via trait-based species sorting, significant patterns of turnover (i.e., beta diversity) in FD, PD (assuming niche conservatism), and TD will be found. In contrast, it may not translate into significant patterns in alpha-level TD, PD, and FD: their values may remain unchanged despite significant species, species-trait, or lineage turnover (*Mouchet et al., 2010*). In addition, addressing both scales of analysis provides complementary information. At the local-level, biotic interactions, environmental filtering, and stochastic processes play major roles in determining (alpha) diversity whereas, at more regional scales, environmental filtering as well as historical and evolutionary processes may largely drive (beta) diversity (*Cavender-Bares et al., 2009*). For instance, it is thought that environmental filtering operates more strongly at the regional scale (*Cornwell, Schwilk & Ackerly, 2006*), while species interactions (e.g., competition) drive local-level assembly patterns (*Cavender-Bares et al., 2009*; *Slingsby & Verboom, 2006*).

In this study, we evaluated how alpha and beta TD, PD, and FD in ant communities across western and central Europe are shaped by environmental and spatial factors, with the aim of understanding the mechanisms that structure communities. We also analyzed trait phylogenetic signals to determine if they helped shape pattern similarity among the three facets of diversity. Ants are a good study system when it comes to examining biodiversity patterns because they are among the most diverse and abundant organisms on earth and perform a great variety of ecological functions that are critical for ecosystems (*Hölldobler & Wilson, 1990*). We compiled data from 349 ant communities that included a total of 155 ant species, and we characterized a set of 10 functional traits that reveal different dimensions of the ant functional niche. We also quantified ant phylogenetic relatedness. Our communities were distributed along different broad environmental gradients (e.g., climate, land-use, and anthropogenic disturbance). This is the first study to analyze the relative contribution of different factors to different facets of biodiversity at two different scales (alpha and beta). We generated the following three hypotheses. First, environmental heterogeneity and space should significantly contribute to variation in TD, PD, and FD at both alpha and beta scales. However, since ants face dispersal limitations
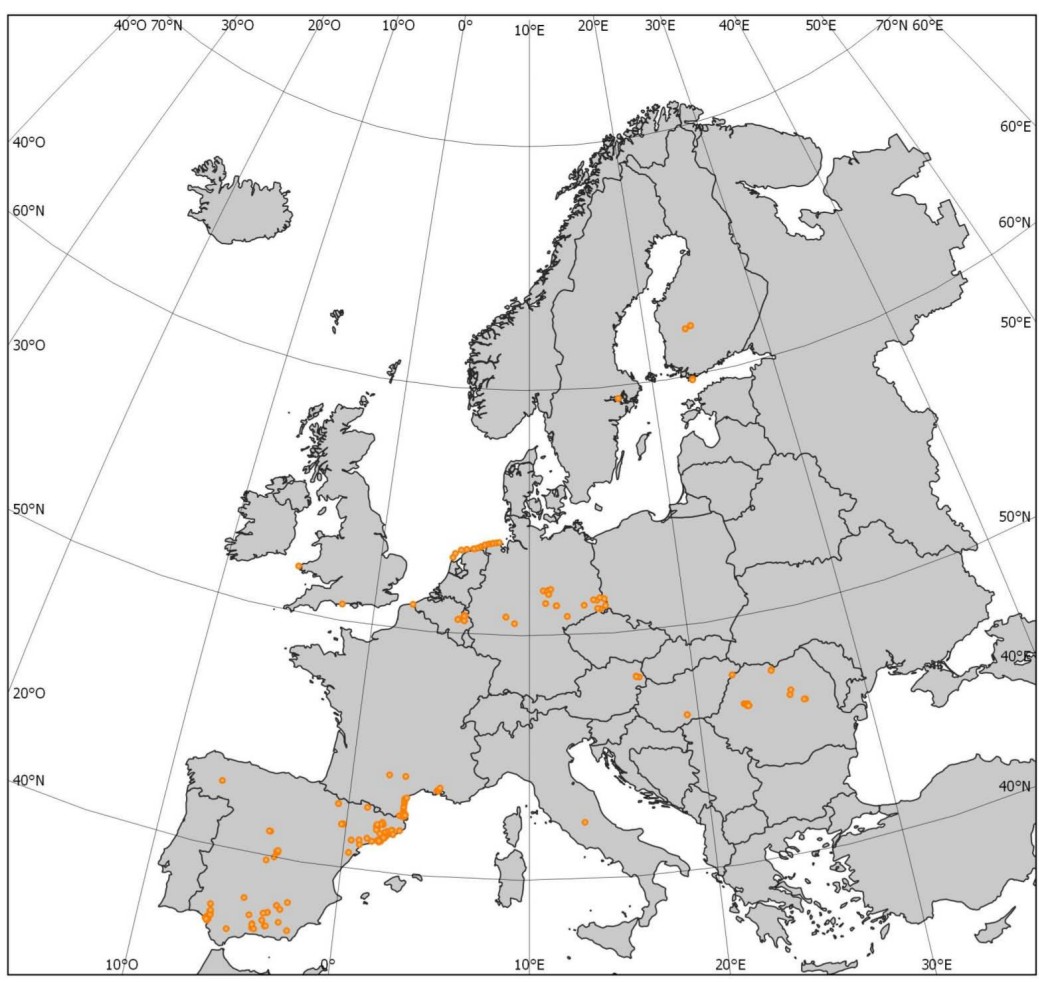

**Figure 1** **Map of the study area showing plot distribution.**

(*Mezger & Pfeiffer, 2011*), we expected spatial factors to make a larger contribution. Second, the different facets of biodiversity should show similar responses along environmental gradients. Third, because environmental filtering is expected to limit community members to those that are preadapted to local conditions, and that are thus functionally similar, environmental factors should influence FD more than TD or PD; in turn, if strong niche conservatism exists, PD and FD should display similar patterns.

## MATERIAL AND METHODS

### Ant community data

Information on the species composition of European ant communities was obtained from primary data collected by the authors and from an exhaustive search of scientific literature that contained species abundance or presence-absence data from single locations. The dataset encompassed 349 sites from eleven different countries (Fig. 1). At these sites, there were a total of 155 ant species (Table S1). As abundance information does not necessarily provide better diversity metrics (*Devictor et al., 2010*), and given that presence-absence

data are more comparable among sites than are abundance data (usually measured in different ways: number of nests, individuals at baits, or individuals in pitfall traps), we focused our analyses on the presence-absence dataset.

### Trait data

We characterized 155 ant species according to 10 traits that determine different dimensions of the ants' functional niches with respect to morphology, life-history, and behavior at both the level of the individual worker and that of the colony (Tables S1 and S2). These traits are considered important in ants because of their influence on ant autoecology and ecosystem functioning (e.g., *Hölldobler & Wilson, 1990*; *Arnan, Cerdá & Retana, 2012*; *Arnan et al., 2013*; *Arnan, Cerdá & Retana, 2014*); furthermore, they strongly respond to environmental gradients (*Arnan, Cerdá & Retana, 2012*; *Arnan et al., 2013*; *Arnan, Cerdá & Retana, 2014*).

### Phylogenetic data

We built a complete phylogeny for the 155 ant species considered (phylogenetic tree provided in Fig. S1). This tree was the product of a super tree derived from a genus-level phylogeny created using a molecular dataset (*Moreau & Bell, 2013*). We then added species to this basal tree by integrating the results of different studies, using a combination of molecular and taxonomic data (Appendix S1). The tree was reconstructed with Mesquite version 3.0 (*Maddison & Maddison, 2014*). For this phylogeny, reliable estimates of branch length and node ages were unavailable. First, to solve the polytomies, we used 'multi2di' from the R (*R Development Core Team, 2010*) package 'phytools'. Second, the tree was ultrametrized applying Grafen's Rho transformation to branch lengths, using the function 'compute.brlen' from the R package 'ape.'

### Environmental gradients

Sites were classified according to their positions along different environmental gradients, which were grouped into two broader gradients:

(a) Climate gradients. Climate data for each site came from the WORLDCLIM database (http://www.worldclim.org/bioclim); rasters of the highest available resolution (30 arc-seconds) were used. We obtained values for four climate variables: mean annual temperature, temperature amplitude (the difference between the maximum and the minimum annual temperatures), annual precipitation, and precipitation seasonality (coefficient of variation of the monthly precipitation level). Previous studies have highlighted the independence of these climate variables (*Arnan, Cerdá & Retana, 2014*).

(b) Land-use gradients. A land-use diversity index was calculated by applying the Simpson index of diversity to the percentage of different land-use categories within a 2-km radius area around the central point of each site. We used seven land-use categories (artificial surfaces, agricultural areas, forests, scrublands, meadows, wetlands, and water bodies), which were obtained from the CORINE land-cover vector database (*Bossard, Feranec & Otahel, 2000*), derived from 25-m resolution satellite data. Similarly, an anthropogenic disturbance index was calculated based on the proportion

of the 2-km radius area around the center of each plot that was occupied by the aforementioned artificial surfaces. We assumed that natural sites within areas with a higher proportion of anthropogenic land-use would be more likely to suffer from anthropogenic disturbance.

All variables were standardized to have a mean of 0 and a variance of 1 (*De Bello et al., 2010*).

## Spatial structure

In order to explore the sites' spatial structure, we generated a set of multiscale principal coordinates of neighbor matrices (PCNM) from the geographic distance matrix using the R package *PCNM*. PCNM eigenfunctions depict a spectral decomposition of the spatial relationships among sites. They are orthogonal sine waves that describe all the spatial scales that can be accommodated in the sampling design (*Dray, Legendre & Peres-Neto, 2006*), such that the first and last axes represent broad- and fine-scale patterns, respectively. Sixty PCNMs were generated.

## Partitioning taxonomic, functional, and phylogenetic diversity

To partition each facet of biodiversity considered (TD, FD, and PD) into alpha and beta components, we used the Rao quadratic entropy index, which provides a standardized methodology for comparing these components within the same mathematical framework (*Pavoine, Dufuor & Chessel, 2004*; *De Bello et al., 2010*; *Devictor et al., 2010*). Moreover, this index makes it possible to calculate functional diversity for combinations of traits, and it can handle quantitative, categorical, and binary traits (e.g., *Rao, 1982*; *Lepš et al., 2006*). Furthermore, its estimates of functional and phylogenetic diversity are relatively independent of taxonomic diversity (e.g., *Mouchet et al., 2010*). We used additive partitioning to break down overall gamma diversity into within (alpha) and among (beta) community diversity. Within each community $k$ with $S$ species, $\alpha$-diversity was calculated using Rao's coefficient of diversity (*Rao, 1982*; *Pavoine, Dufuor & Chessel, 2004*) modified for presence-absence data:

$$\alpha \, \text{Rao}_{(k)} = \sum_{i=1}^{S} \sum_{j=1}^{S} d_{ij}$$

where $d_{ij}$ is the distance between species $i$ and $j$, which can be taxonomic, functional, or phylogenetic. This index represents the expected dissimilarity between two individuals of different species chosen randomly from the community. Between communities $k$ and $l$, $\beta$-diversity was computed using the Rao's dissimilarity index (*Rao, 1982*; *Pavoine, Dufuor & Chessel, 2004*), which is the expected distance between two individuals of different species chosen randomly from two distinct communities:

$$\beta \, \text{Rao}_{\text{pairwise}(k,l)} = (\gamma_{(k+l)} - \bar{\alpha}_{(k,l)})/\gamma_{(k+l)}$$

where $\gamma_{k+1}$ is the gamma diversity of the pair of communities (calculated with the same equation as for alpha diversity, but taking into account all the species included in the

two communities) and $\bar{\alpha}_{(k,l)}$ is the mean $\alpha$-diversity of the two communities. Prior to performing the calculations, we applied Jost's correction (*Jost, 2007*) to $\gamma$ and $\alpha$ to properly quantify $\beta$-diversity independently of $\alpha$-diversity (*De Bello et al., 2010*). To carry out these calculations, we used the function 'rao' (*De Bello et al., 2010*) in R.

To calculate the Rao quadratic entropy index, different distance measures were used depending on the facet of diversity considered. Taxonomic distances between species were measured as $d_{ij} = 1$ when $i \neq j$, and $d_{ij} = 0$ when $i = j$. To compute functional distances between species, we first conducted a principal component analysis (PCA) on the standardized (mean = 0, SD = 1) trait data to correct for dominance in the distance matrix by highly correlated traits (*Devictor et al., 2010*; *Purschke et al., 2013*). The resulting PCA axes were used to calculate Euclidean distances. Phylogenetic distances between species were measured with the cophenetic distances from the phylogenetic tree. We scaled all distances between 0 and 1 by dividing each type of distance by its maximum value in order to make taxonomic, functional, and phylogenetic distances comparable.

## Statistical analyses

Moran's I and Mantel tests were used to test for spatial autocorrelation between the alpha and beta components of TD, PD, and FD, respectively.

We used redundancy analysis (RDA) with variation partitioning (*Borcard, Legendre & Drapeau, 1992*) to assess the relative influence of environmental and spatial factors, alone and combined, on alpha- and beta-level variation in TD, PD, and FD. We partitioned the variation into multiple components: a pure environmental component, a pure spatial component, a spatially structured environmental component, and residual variation. Forward selection (*Blanchet, Legendre & Borcard, 2008*) was used for each set of environmental and spatial variables to select only those variables that significantly explained variation in the dependent variables ($p < 0.05$, after 999 random permutations). Only the selected variables were used in variation partitioning. $R^2$ values adjusted for the number of sites and explanatory variables were used throughout because they provided corrected estimates of explained variation ($R^2_{adj}$; *Peres-Neto et al., 2006*). Monte Carlo permutation tests (9,999 permutations) were used to calculate the significance levels of the different components. Since the beta diversity indices estimated with the RaoQ index were dissimilarity matrices, they cannot be used directly as response variables in this type of variation-partitioning framework. Therefore, prior to conducting the RDA analysis and variation partitioning, we transformed the dissimilarity matrices into data frames by conducting a principal coordinate analysis (PCoA) on the dissimilarity matrices; we then used the scores of the significant axes as the representative values for each community (*Legendre & Anderson, 1999*). Mixing traits in the functional diversity index as we did could have resulted in FD demonstrating a neutral response to gradients; this does not occur when individual traits are used. To address this problem, we also conducted analyses of FD for each trait separately.

A relationship between FD and PD could be explained by significant phylogenetic signals in the traits used to calculate FD. We therefore tested for their presence using Pagel's

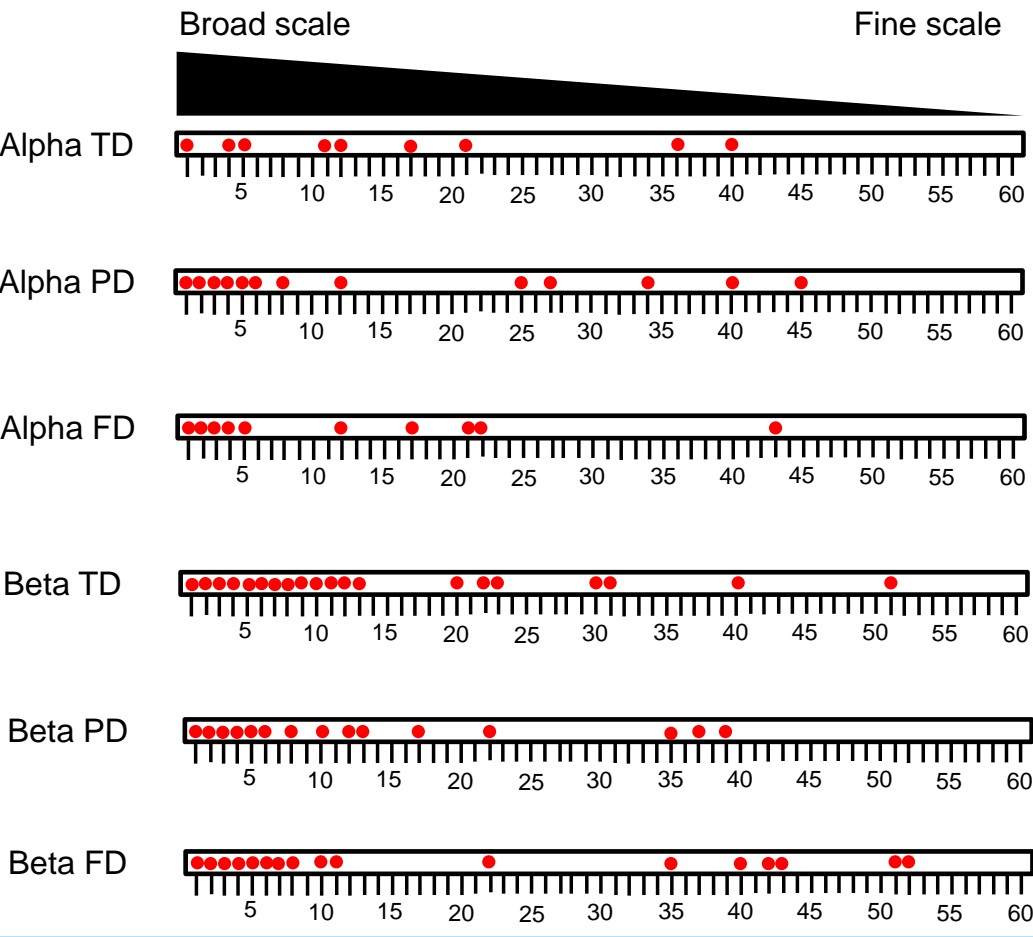

**Figure 2 Correlograms for the alpha and beta components of the taxonomic, phylogenetic and functional diversity.** We used the Moran's I and Mantel tests for alpha and beta diversity, respectively. Black circles indicate significant correlation ($p < 0.05$).

$\lambda$ test (*Pagel, 1999*), which assumes a Brownian motion (BM) model of trait evolution. To test for statistical significance, we used a likelihood ratio test approximated by a chi-squared distribution to compare the negative log likelihood obtained when there is no signal (i.e., using the tree transformed $\lambda = 0$) to that estimated from the original topology.

All analyses were conducted in R using the packfor, PCNM, vegan, ade4, and Geiger packages.

## RESULTS

We found strong autocorrelations between both the alpha and beta components in the three facets of diversity (Fig. 2). According to our first hypothesis, our RDA analyses revealed that both environmental and spatial factors significantly contributed to variation in alpha- and beta-level TD, PD, and FD (Fig. 3). Furthermore, we found that spatial factors made a much larger contribution than environmental factors for all dependent variables (Fig. 3), which indicates that ant communities within our study area were

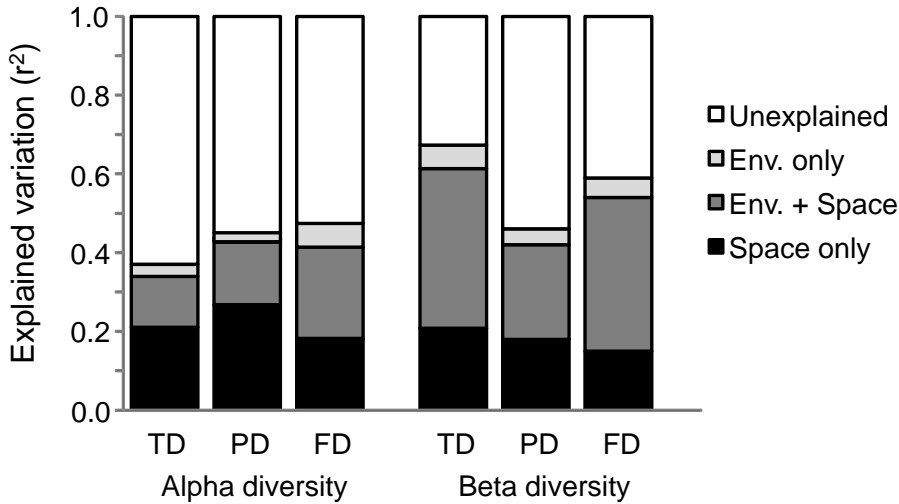

**Figure 3 Partitioning of variation in alpha- and beta-level taxonomic (TD), phylogenetic (PD), and functional (FD) diversity.** The figure depicts the adjusted unique contribution ($R^2$) of environmental factors (light gray), spatial factors (black), spatially structured environmental factors (dark gray), and unexplained variation (white), as calculated in the RDA analyses. Tests of significance for the environmental and spatial factors were all significant ($p < 0.05$).

strongly structured by space. The PCNMs retained in the alpha- and beta-level models of spatial structure were mostly broad in scale (Fig. 4).

Although environmental factors made a relatively smaller contribution, they nonetheless appear to play an important role in structuring ant diversity. Interestingly, the different facets of diversity mostly responded to similar environmental gradients in the same way (Table 1), which gives support to our second hypothesis. In the alpha-level RDAs, two of the six environmental variables (mean annual temperature and anthropogenic disturbance) were retained in the TD, PD, and FD models, which highlight the role of these two factors in determining ant diversity. In particular, warmer and less disturbed sites had higher levels of all three types of diversity. Mean annual temperature explained most of the variation in PD and FD; in the case of TD, anthropogenic disturbance explained an equivalent amount of variation. TD and FD were positively correlated with temperature amplitude, and TD and FD were negatively correlated with precipitation seasonality and annual precipitation, respectively (Table 1). No facet of diversity was influenced by land-use diversity. In the beta-level RDAs, four of the six environmental variables (mean annual temperature, temperature amplitude, precipitation seasonality, and anthropogenic disturbance) were retained in the TD, PD, and FD models (Table 1). More specifically, the greater the distance among these environmental variables, the higher the turnover in TD, PD, and FD. Once again, mean annual temperature explained most of the variation. Turnover in TD was mediated by differences in annual precipitation, while turnover in FD was mediated by annual precipitation and land-use diversity (Table 1).

At the alpha scale, environmental factors accounted for only a small fraction of the total variation in TD (3%), PD (1%), and FD (6%). Spatial factors had greater explanatory ability (21, 27, and 21%, respectively). Spatially structured environmental factors

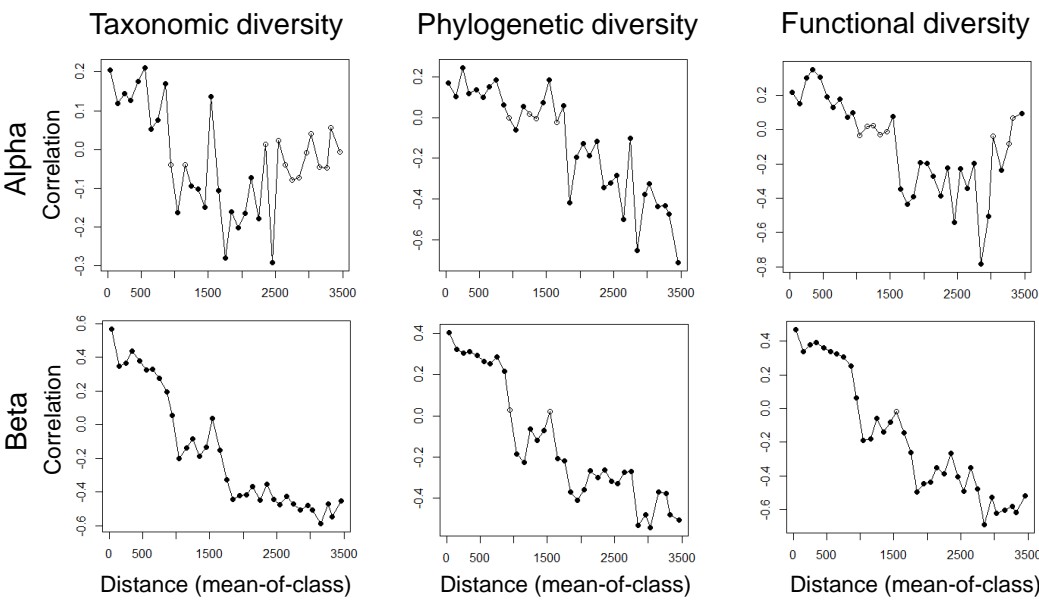

**Figure 4 Selected PCNMs from the 60 PCNMs that exhibit a positive spatial correlation for the alpha and beta components of taxonomic (TD), phylogenetic (PD) and functional (FD) diversity.** The selected PCNMs from forward selection analyses are represented by red circles.

**Table 1 Variables retained, adjusted R-squared, and significance values from the forward-selected models examining the effect of environmental factors on alpha- and beta-level taxonomic, phylogenetic, and functional diversity.** The directions of the significant relationships are depicted in brackets.

| | Alpha diversity | | | Beta diversity | | |
|---|---|---|---|---|---|---|
| | Taxonomic diversity | Phylogenetic diversity | Functional diversity | Taxonomic diversity | Phylogenetic diversity | Functional diversity |
| Mean annual temperature | (+) 0.06*** | (+) 0.15*** | (+) 0.18*** | (+) 0.35*** | (+) 0.24*** | (+) 0.35*** |
| Temperature amplitude | (+) 0.01* | | (+) 0.09*** | (+) 0.03*** | (+) 0.02*** | (+) 0.07*** |
| Annual precipitation | | | (−) 0.01* | | (+) 0.01*** | (+) 0.005* |
| Precipitation seasonality | (−) 0.02** | | | (+) 0.06*** | (+) 0.01** | (+) 0.02*** |
| Land-use diversity index | | | | | | (+) 0.01** |
| Anthropogenic disturbance index | (−) 0.06*** | (−) 0.02** | (−) 0.02** | (+) 0.02*** | (+) 0.01** | (+) 0.005* |
| Full-model adjusted $R^2$ | 0.15 | 0.17 | 0.29 | 0.47 | 0.28 | 0.45 |

**Notes.**
* $P < 0.05$.
** $p < 0.01$.
*** $p < 0.001$.

explained 13, 16, and 23% of the total variation in TD, PD, and FD, respectively (Fig. 3). Environmental factors (alone, and in tandem with space) made a greater contribution to FD than to TD and PD. These results are in agreement with our third hypothesis. Like at the alpha scale, at the beta scale, environmental factors accounted for only a small fraction of the total variation in TD (6%), PD (4%), and FD (5%). Spatial factors explained 21, 18, and 15% of the variation in TD, PD, and FD, respectively. In contrast, spatially structured

environmental factors had the greatest explanatory ability (TD: 41%, PD: 24%, and FD: 39%, respectively). Environmental factors (alone, and in tandem with space) made a greater contribution to TD and FD than to PD. Overall, they explained much more beta- than alpha-level variation in TD, PD, and FD, especially in the case of TD and FD (Fig. 3).

When the functional traits were examined separately (Table S3 and Fig. S2), it was clear that FD was shaped by individual trait responses at both the alpha and beta scales. Traits responded differently (positively or negatively) to the various environmental gradients and made different contributions to the environmental and spatial components. Indeed, the global FD pattern did not seem to be driven by any particular trait, since no single trait showed the same pattern as global FD. Furthermore, individual functional responses did not mirror TD and FD patterns at either the alpha or the beta scale. However, all the functional traits had significant phylogenetic signals (Table S4), which is evidence (albeit not definitive) for niche conservatism. The values of Pagel's λ ranged from 0.78 for polydomy to 1 for diet (seeds, in particular) and foraging strategy, which demonstrates that phylogenetic signals were strong for most of the functional traits.

## DISCUSSION

In accordance with our first hypothesis, we found that environmental factors and spatial factors separately influence alpha- and beta-level TD, PD, and FD. In particular, it seems that climate and human-modified landscapes shape the different facets of ant diversity in western and central Europe. More interestingly, spatial factors played a large role, which suggests that dispersal limitations have a strong effect on ant community structure at the different diversity levels. However, the strength of the role of spatial factors on shaping the different facets of ant diversity may reflect the omission of certain spatially structured environmental variables (Jones et al., 2008). We refute this interpretation for several reasons. First, we examined a wide range of key gradients, including the main climate and human-modified landscape gradients in western and central Europe. Other studies have found that similar gradients have an influence on the structure of animal communities (Meynard et al., 2011), including ant communities (Arnan, Cerdá & Retana, 2012; Arnan, Cerdá & Retana, 2014). Second, there were strong spatial autocorrelations in all the diversity metrics (Fig. 1), and we found that alpha- and beta-level spatial structure in all the diversity facets was best explained by broad-scale PCNMs (Fig. 4). Third, our study area encompassed much of western and central Europe, which contains a great diversity of habitats, and even spanned across some large mountain ranges. Fourth, ant gynes seem to face dispersal limitations as a general rule (Mezger & Pfeiffer, 2011) because ants are small and their alates are often poor dispersers. Altogether, this evidence suggests that, given the presence of mountain ranges, habitat diversity, and other obstacles, dispersal limitations strongly structure ant communities across Europe at all diversity levels. In this respect, our results agree with those of Meynard et al. (2011), who found that spatial structure played a preponderant role in structuring alpha- and beta-level TD, PD, and FD in French bird communities. Dispersal limitations are much more severe in ants than in birds, and, accordingly, our study found a stronger effect of spatial factors on diversity.

Although we found that ant diversity patterns in western and central Europe are determined by spatial and environmental parameters, our results also highlight a large portion of unexplained variation in alpha and beta diversity patterns. This fact suggests that other processes are at work in determining ant diversity patterns. For instance, ant diversity patterns across Europe might also be determined by stochastic mechanisms, assuming that population dynamics do not depend on environmental characteristics and are primarily driven by ecological drift and dispersal (*Hubbell, 2001*). Also, the omission of non-spatially structured biological or environmental variation might also account for part of the unexplained variance. For instance, it is known the role of competitive interactions in structuring ant diversity at the local scale (*Cerdá, Arnan & Retana, 2013*); the fact that the portion of unexplained variance is higher for the alpha than for the beta diversity (Fig. 3), might suggest that variation in competitive interactions along the gradients might account for part of the unexplained variance in ant diversity patterns.

We also found largely support for our second hypothesis: TD, PD, and FD in ant communities demonstrated parallel responses along most of the environmental gradients in western and central Europe. Given that spatially structured environmental factors had strong effects (Fig. 1), a possible explanation is that topography-related dispersal limitations affect particular functional groups and/or lineages, and consequently, particular species. It is worth mentioning here that we found strong phylogenetic signals in the traits we examined (Table S3), which might explain why PD and FD showed highly similar responses. Once again, our results corroborated those of *Meynard et al. (2011)*, the only other study conducted thus far that had similar aims and a comparably large spatial scale. The authors found general support for the idea that hypotheses generated for local and regional TD can be extended to PD and FD. Conversely, *Bernard-Verdier et al. (2013)* observed no congruence among alpha- and beta-level TD, PD, and FD along local gradients of soil type and resource availability. *Purschke et al. (2013)* found similar incongruence in a plant community over the course of plant succession. A recent study that analyzed the FD of plants along a latitudinal gradient in the New World found that patterns of alpha-, beta-, and gamma-level diversity failed to match any one theory of species diversity (*Lamanna et al., 2014*). A global comparison of mammalian diversity found that TD, PD, and FD are somewhat related and concluded that any mismatches were attributable to environmental factors (*Safi et al., 2011*).

However, we also found some environment-mediated mismatches, which might be due to assorted environmental drivers differentially filtering the different facets of diversity (*Safi et al., 2011*; *Hermant et al., 2012*). For instance, diversity patterns varied along the annual precipitation gradient. Precipitation usually influences primary productivity and resource availability (*Leith & Whittaker, 1975*). The lower functional diversity in wetter areas might stem from relaxed local competition: higher levels of primary productivity could mean less competition for resources and thus lower rates of species extinction, ultimately resulting in functional redundancy (*Pavoine & Bonsall, 2011*). Consequently, functional turnover should be much higher than taxonomic turnover along the precipitation gradient, which is what we observed. Moreover, the degree of

niche conservatism might vary for different environmental gradients, which might explain why patterns among the three facets of diversity differed in some gradients. However, this speculation remains outside the scope of our paper.

In our study, the most striking environmental gradient was a climate gradient, along which mean annual temperature varied: it influenced all three facets of diversity and explained much more variation than the other environmental factors. This finding highlights the role of temperature as one of the main drivers of biodiversity, which supports concerns about the effects of climate change on species distributions (*Dunn et al., 2009*; *Jenkins et al., 2011*), as well as related ecosystem services and evolutionary responses. The negative effect that anthropogenic disturbance had on all three facets of diversity at both alpha and beta scales—triggering diversity turnover—should also be underscored. Furthermore, land-use diversity affected functional turnover. These findings are noteworthy given concerns about the functional consequences of current biodiversity losses (*Loreau et al., 2001*), especially those mediated by human-driven changes (*Foley et al., 2005*); indeed, the most important driver of declining biodiversity is changes in land use (*Sala et al., 2000*). In accordance with our third hypothesis, FD responded to more environmental gradients than either TD or PD did. Furthermore, when all three facets of diversity significantly responded to the same environmental gradient, relatively more variation in FD was explained. However, our single-trait analyses suggest that this finding might be contingent on the trait examined. At any rate, the large contribution of environmental gradients to the multi-trait FD index seems somewhat obvious, because a species' traits clearly determine whether it will successfully pass through an environmental filter (*Pavoine & Bonsall, 2011*) and consequently are the underlying force shaping functional composition. Interestingly, environmental factors explained a similar amount of beta-level variation in TD and FD, which suggests that strong environmental filtering is operating along these gradients (*Mouchet et al., 2010*). If species sorting is weak, we would not expect to see major changes in functional traits, i.e., we would not expect high functional turnover even if species turnover occurred. In contrast, if there is strong species sorting along environmental gradients, we would expect both species and functional turnover (*Mouchet et al., 2010*).

Notably, environmental factors explained more alpha- and beta-level variation in FD than in PD. In birds, *Meynard et al. (2011)* found the opposite pattern; they attributed this finding to either overlooking some of the relevant gradients that affect FD, using poorly chosen life-history traits to measure FD, or the fact that PD is simply a more integrative proxy than FD when analyzing a given subset of life-history traits. In this study, we refute the general assumption that PD is a more integrative measure of FD (*Cadotte et al., 2009*; *Meynard et al., 2011*), and we give empirical support to those authors who have questioned this assumption from a theoretical standpoint (*Losos, 2008*; *Cavender-Bares et al., 2009*; *Mouquet et al., 2012*). Phylogenetic diversity does not appear to be a good proxy for FD when a large number of well-chosen, diverse functional traits are used. We selected morphological, life-history, and behavioral traits related to resource exploitation, reproduction, and social structure; these traits have been demonstrated to be strongly

related to the abiotic environment (*Arnan et al., 2013*; *Arnan, Cerdá & Retana, 2014*) and may therefore reflect effective local adaptation. Our results lend credence to the idea that FD is a better indicator of community assembly processes than PD (*Díaz et al., 2007*).

We also found that environmental factors had a stronger effect on TD, PD, and FD at the beta level than at the alpha level, which supports the idea that environmental filtering is stronger at the regional scale (*Cornwell, Schwilk & Ackerly, 2006*). Other kinds of mechanisms, such as species interactions (e.g., competition or facilitation), might have larger effects at the local scale than at the regional scale (*Cavender-Bares et al., 2009*; *Slingsby & Verboom, 2006*). Moreover, beta diversity responded more similarly to environmental gradients than did alpha diversity. This finding concurs with the results of some past work (*Devictor et al., 2010*; *Bernard-Verdier et al., 2013*), which found relatively greater congruence among different facets of beta-level vs. alpha-level diversity.

Finally, we should assume that our study might have some limitations, mainly related to the fact that the communities we used are not evenly distributed across the spatial coverage of this study, and they account for a subset of species known to occur in western and central Europe. However, our study encompasses the most comprehensive dataset on ant communities in Europe. Although our communities do not display a regular spatial distribution, they encompassed most of the range that the environmental variables take across western and central Europe; moreover, the species we found in this study are the most common species of the region. All this suggest that the results from our sampled communities are representative of the patterns of ant diversity in western and central Europe.

## CONCLUSIONS

By using variation-partitioning analyses, we have demonstrated that ant diversity patterns in western and central Europe, whether TD, PD, or FD, are driven by both environmental determinism and dispersal limitations, with the latter playing a more prominent role. The strong autocorrelations that we found in our diversity data, along with the potent effects of dispersal limitations, underscore that TD, PD, and FD are highly heterogeneous in western and central Europe. This finding implies that western and central European ant communities are taxonomically, phylogenetically, and functionally clustered (*Zupan et al., 2014*), which might be the result of historical and/or evolutionary forces. For instance, the study area is composed of highly diverse biogeographic regions, which might display different rates of trait evolution and speciation (e.g., *Weir & Schluter, 2007*; *Cooper & Purvis, 2010*). Recent massive diversification events (*Slingsby & Verboom, 2006*), and the different historical disturbance regimes at the origin of current-day European landscapes (*Schelhaas, Nabuurs & Schuck, 2003*) might contribute to this heterogeneity in ant diversity. Our findings also highlight that the different facets of diversity are fairly equivalent because they demonstrate similar patterns along environmental gradients. However, environmental factors explained the most variation in FD, which reflects the strong effect that species sorting (i.e., environmental filtering) has on these patterns. Furthermore, functional traits had strong phylogenetic signals, which suggests that niche conservatism might account for the parallel patterns displayed by FD and PD and, therefore, also have

relevance for diversity patterns in general. It is clear that incorporating phylogenetic relationships and functional ecology into analyses of ecological patterns allowed us to draw stronger conclusions regarding the mechanisms that underlie macroecological patterns at different spatial scales (*Webb et al., 2002*; *Cavender-Bares et al., 2009*). There is thus a definite need to integrate the information furnished by different facets of diversity to better understand the assembly rules responsible for current global patterns of biodiversity.

## ACKNOWLEDGEMENTS

We are very grateful to Alejandro González-Voyer and Iván Gómez-Mestre for their help with the phylogeny, Bernhard Seifert and Laszlo Gallé for providing their data on central European ant communities and Jessica Pearce-Duvet for her English editing services.

### Funding

This study was partly funded by the Spanish 'Ministerio de Economía y Competitividad' and FEDER (project CGL2012-36181 to XC) and the 'Ministerio de Ciencia e Innovación' (project Consolider MONTES, CSD 2008-00040 to JR). XA was supported by the Conselho Nacional de Desenvolvimento Científico e Tecnológico of Brazil (CNPq PDS-167533/2013-4). The funders had no role in study design, data collection and analysis, decision to publish, or preparation of the manuscript.

### Grant Disclosures

The following grant information was disclosed by the authors:
Spanish 'Ministerio de Economía y Competitividad' and FEDER: CGL2012-36181.
Ministerio de Ciencia e Innovación: 2008-00040.
Conselho Nacional de Desenvolvimento Científico e Tecnológico of Brazil: PDS-167533/2013-4.

### Competing Interests

The authors declare there are no competing interests.

### Author Contributions

- Xavier Arnan conceived and designed the experiments, performed the experiments, analyzed the data, contributed reagents/materials/analysis tools, wrote the paper, prepared figures and/or tables.
- Xim Cerdá performed the experiments, reviewed drafts of the paper.
- Javier Retana conceived and designed the experiments, performed the experiments, reviewed drafts of the paper.

### Supplemental Information

Supplemental information for this article can be found online at http://dx.doi.org/10.7717/peerj.1241#supplemental-information.

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
