# Peer review of "Partitioning the impact of environment and spatial structure on alpha and beta components of taxonomic, functional, and phylogenetic diversity in European ants"

_PeerJ, doi:10.7717/peerj.1241_

## Round 0.1 · original submission · Minor Revisions

· Academic Editor

Minor Revisions

Dear Xavier, please find comments on your submission, both reviewers found it interesting and a significant contribution. Please try and clarify parts of the manuscript where the reviewers have made comments.

Reviewer 1 ·

Basic reporting

This article is written in a sound style and good English, which apparently has been checked and edited by native speaker.
The introductory part seems to be quite exhaustive given the topic of the paper, although I miss a note on some of the other concepts of functional grouping of ants that has been globaly used in scientiffic literature (e.g. Andersen 1997).

There is some discordance of the manuscript with the journals instructions, some of which I consider quite problematic:
In the reference part, all references should include full list of authors which is not the case in several cases - see lines 467, 472, 474, 484, 493, 498, 504, 524, 555, 572.
L. 274 - Figure 2 is mentioned. However, there is no such figure included.
In Abstract and in Discussion (e.g. L. 297) the authors state, that their data are related with W. Europe. However, their dataset originated also from sites in Austria, Hungary and Romania, which are not exactly Western countries.
Regarding the Result section I really see no reason why supplementary Figures S1, S3 and S4 should not be an ordinary part of this section.
In Introduction and Results the different facets of biodiversity are mentioned in the form of acronyms. However in the Discussion they all are mentioned in full form. I suggest to use them in an unified manner (I personally prefer the full form).
In Figure 1 the grey scale is a bit confusing.

Experimental design

This is probably the first study analyzing the effect of different environmental and spatial factors on the diversity of ants across different facets and levels of biodiversity, which is so far new in the European context. However some data background is needed to fully undertsnad the methodological approach.
Especially in the supplementary files we see no information on how the functional and phylogenetic traits were attributed to the particular ant species, nore see we the list of the 155 species considered.
The phylogenetic tree of the 155 ant species is mentioned as a part of the supplementary Figure S2, however, the Figure S2 is in fact not included.

Validity of the findings

There is no conclusion chapter included in the manuscript. Since the discussion chapter is quite extensive, it would be quite usefull to have a brief and general conclusion added. It will also allow the authors to add some notes on the applications of their work.
I wonder whether the set of 155 ant species is exhaustive and representative enough to assess such broad landscape and ecological gradients as presented, taking into account that e.g. in Spain the species richness of ants is not lower than 318 spp (antmaps.org). Also the collection sites are anything but evenly distributed across the region, which may have a profound effect on the datased and the ecological background. I would really like to see some of the authors thoughs on this topic in the discussion.
I also see no proper discussion on the large portion of unexplained variation in alpha and beta diversity as seen in Figure 1.

Additional comments

This is a nice study. Maybe to much unambiguous (definite) based on the used dataset, but still novel and usefull. Please add the missing parts and try to discuss the representativeness of your assessments in the manuscript before it can be accepted for publication.

·

Basic reporting

The manuscript meets all requirements.

Overall this is a very ambitious project that drives at very important ecological concepts and is well supported through analyses and data. This manuscript is well deserving of publication with some minor edits/clarifications. I should note that I am not completely familiar with all of the analyses in this manuscript but where I was unfamiliar, a check of the literature seems to suggest the approaches used are appropriate.

Experimental design

Might be a role for ground truthing the ant taxonomy (voucher specimens) of random studies included in the meta-analysis to determine whether misidentifications could be a significant factor affecting estimates of beta diversity. I note this generally as something that might be considered in the future and do not think it would significantly affect the results. I only note that ant taxonomy is more fraught with misidentifications that one might expect for other taxa.

Validity of the findings

The manuscript clearly lays out three hypotheses which are methodically tested.

The first hypothesis suggests that, “environmental heterogeneity and space should significantly contribute to variation in TD, PD, and FD at both alpha and beta scales.” This is well tested and supported by the analyses.

The second hypothesis predicts that “the different facets of biodiversity should show similar patterns along environmental gradients.” It would help if this hypothesis was more precisely stated with respect to the more precise meaning of ‘pattern.’ In the Discussion the resolution of this hypothesis is somewhat ambiguous with an opening statement that the hypothesis is supported (Line 317) but then some apparent contradiction (Line 344-346). The importance of this latter conflict is not clear.

The third hypothesis, predicts that, “because environmental filtering is expected to limit community members to those that are preadapted to local conditions, and that are thus functionally similar, environmental factors should influence FD more than TD or PD; in turn, if strong niche conservatism exists, PD and TD should display similar patterns.” The stronger role for FD is apparent and discussed but I am not sure how well the relationship between PD and TD are discussed. PD is related in a few instances to FD, but not really discussed in the context of TD. However, given the complexity of this work, perhaps I am missing this

Additional comments

It would help if the results section was framed by the hypotheses. This is a complex study looking at multiple hypotheses, multiple biodiversity indices across multiple gradients. Placing the results within the structure of the hypotheses will clarify the data being presented.

Line 95 Use ‘local-level’ rather than ‘community scale’ for consistency and to avoid confusion with the term ‘community.’
Line 149-150. Minor edit. Place period inside single quotation mark.
Line 163: what were the plots?
Line 171: reference?
Line 299: “…the strength of this role…” Exactly what role is being referenced? I believe you mean dispersal limitations but it is not clear.
Line 319: needs a data reference. This should be noted throughout the Discussion.
Line 322: “ signals in the traits we examined,…”  “ signals in the functional traits we examined,…

---

## Round 0.2 · accepted · Accept

· Academic Editor

Accept

Hi Xavier, thank you for your considerations and changes to the reviewer comments. This is a neat paper!